

# Quantile-dependent expressivity of plasma adiponectin concentrations may explain its sex-specific heritability, gene-environment interactions, and genotype-specific response to postprandial lipemia

Paul T. Williams

Molecular Biophysics & Integrated Bioimaging, Lawrence Berkeley National Laboratory, Berkeley, CA, United States of America

## ABSTRACT

**Background**. "Quantile-dependent expressivity" occurs when the effect size of a genetic variant depends upon whether the phenotype (e.g. adiponectin) is high or low relative to its distribution. We have previously shown that the heritability ($h^2$) of adiposity, lipoproteins, postprandial lipemia, pulmonary function, and coffee and alcohol consumption are quantile-specific. Whether adiponectin heritability is quantile specific remains to be determined.

**Methods**. Plasma adiponectin concentrations from 4,182 offspring-parent pairs and 1,662 sibships from the Framingham Heart Study were analyzed. Quantile-specific heritability from offspring-parent ($\beta_{OP}$, $h^2 = 2\beta_{OP}/(1 + r_{spouse})$) and full-sib regression slopes ($\beta_{FS}$, $h^2 = \{(1 + 8r_{spouse}\beta_{FS})^{0.05}\text{-}1\}/(2r_{spouse})$) were robustly estimated by quantile regression with nonparametric significance assigned from 1,000 bootstrap samples.

**Results**. Quantile-specific $h^2$ ($\pm$ SE) increased with increasing percentiles of the offspring's age- and sex-adjusted adiponectin distribution when estimated from $\beta_{OP}$ ($P_{trend} = 2.2 \times 10^{-6}$): $0.30 \pm 0.03$ at the 10th, $0.33 \pm 0.04$ at the 25th, $0.43 \pm 0.04$ at the 50th, $0.55 \pm 0.05$ at the 75th, and $0.57 \pm 0.08$ at the 90th percentile, and when estimated from $\beta_{FS}$ ($P_{trend} = 7.6 \times 10^{-7}$): $0.42 \pm 0.03$ at the 10th, $0.44 \pm 0.04$ at the 25th, $0.56 \pm 0.05$ at the 50th, $0.73 \pm 0.08$ at the 75th, and $0.79 \pm 0.11$ at the 90th percentile. Consistent with quantile-dependent expressivity, adiponectin's: (1) heritability was greater in women in accordance with their higher adiponection concentrations; (2) relationships to *ADIPOQ* polymorphisms were modified by adiposity in accordance with its adiponectin-lowering effect; (3) response to rosiglitazone was predicted by the 45T> G *ADIPOQ* polymorphism; (4) difference by *ADIPOQ* haplotypes increased linearly with increasing postprandial adiponectin concentrations.

**Conclusion**. Adiponectin heritability is quantile dependent, which may explain sex-specific heritability, gene-environment and gene-drug interactions, and postprandial response by haplotypes.

Corresponding author
Paul T. Williams,
1742spyglass@comcast.net,
ptwilliams@lbl.gov

## INTRODUCTION

Adiponectin is a 30 kDa circulating adipocyte-derived protein that is a potent insulin sensitizer that regulates energy homeostasis and glucose tolerance in muscle and liver (*Swarbrick & Havel, 2008*). Low adiponectin concentrations are associated with insulin resistance, type 2 diabetes mellitus (T2DM), coronary artery disease, lipodystrophy, nonalcoholic hepatic steatosis, and essential hypertension, and they precede the development of insulin resistance and myocardial infarction (*Swarbrick & Havel, 2008*). Meta-analysis showed that low plasma adiponectin concentrations predicted increased T2DM risk in 14,598 subjects from 13 prospective studies (*Li et al., 2009*). Paradoxically, prospective studies also find that high adiponectin concentrations is a risk factor for all-cause and cardiovascular mortality (*Menzaghi & Trischitta, 2018*).

Twenty published estimates of adiponectin heritability show its plasma concentrations to be highly heritable (i.e., $h^2 = 0.39$ (*Lindsay et al., 2003*; *Liu et al., 2008*), 0.42 (*Hicks et al., 2007*; *Comuzzie et al., 2001*), 0.47 (*Vaughan et al., 2015*) 0.48 (*Chuang et al., 2004*; *Dosaev, Prakash & Livshits, 2014*), 0.55 (*Pollin et al., 2005*; *Henneman et al., 2010*), 0.58 (*Menzaghi et al., 2010*), 0.62 (*Al-Daghri et al., 2011*), 0.64 (*Guo et al., 2006*), 0.67 (*Ling, Waterworth et al., 2009*), 0.68 (*Ling, Waterworth et al., 2009*), 0.70 (*Chuang et al., 2004*; *Vasseur et al., 2002*), 0.71 (*Guo et al., 2006*), 0.79 (*Menzaghi et al., 2010*), 0.88 (*Cesari et al., 2007*), 0.93 (*Butte et al., 2005*)). None report any difference in heritability between sexes. All but two studies (*Hicks et al., 2007*; *Comuzzie et al., 2001*) used adiponectin concentrations that were logarithmically (*Lindsay et al., 2003*; *Liu et al., 2008*; *Chuang et al., 2004*; *Dosaev, Prakash & Livshits, 2014*; *Pollin et al., 2005*; *Henneman et al., 2010*; *Menzaghi et al., 2010*; *Al-Daghri et al., 2011*; *Guo et al., 2006*; *Vasseur et al., 2002*; *Cesari et al., 2007*; *Butte et al., 2005*) or cube-root transformed (*Vaughan et al., 2015*; *Ling, Waterworth et al., 2009*). The variation in heritability estimates across reports is likely the result of small sample size, different statistical methodologies, differences between twin- and pedigree-based estimates, and population heterogeneity.

"Quantile-dependent expressivity" is said to occur when the phenotypic expression of a gene depends upon the percentile of the phenotype, i.e., whether the trait (e.g., adiponectin) is high or low relative to its distribution (*Williams, 2012*). This is in contrast to the traditional estimate of a genetic effect size that is assumed to be constant across all population percentiles. Quantile-dependent expressivity has been demonstrated for adiposity (*Williams, 2012*; *Williams, 2020c*), lipoproteins (*Williams, 2012*; *Williams, 2020e*; *Williams, 2020a*), pulmonary function (*Williams, 2020g*), coffee intake (*Williams, 2020d*), and alcohol intake (*Williams, 2020f*). Moreover, the genetic effect sizes of single nucleotide polymorphisms (SNPs) affecting triglycerides have been shown to increase and decrease within individuals in accordance with increasing and decreasing postprandial triglyceride concentrations, consistent with quantile-dependent expressivity (*Williams, 2020b*).

An important consequence of quantile-dependent expressivity is that the selection of subjects for characteristics that distinguish high vs. low phenotypes can yield different genetic effects (*Williams, 2012*; *Williams, 2020e*). Adiponectin concentrations are greater in women than men (*Liu et al., 2008*; *Hicks et al., 2007*; *Comuzzie et al., 2001*; *Vaughan*

*et al., 2015*; *Chuang et al., 2004*; *Dosaev, Prakash & Livshits, 2014*; *Henneman et al., 2010*; *Guo et al., 2006*; *Ling, Waterworth et al., 2009*; *Vasseur et al., 2002*; *Cesari et al., 2007*; *Berra et al., 2006*), increase with rosiglitazone treatment (*Kang et al., 2005*), increase during postprandial lipemia (*Musso et al., 2008*), and decrease with adiposity (*De Luis et al., 2020*; *De Luis et al., 2018*; *De Luis et al., 2019*; *Corbi et al., 2019*; *Divella et al., 2017*; *Garcia-Garcia et al., 2014*; *Berthier et al., 2005*; *Aller et al., 2019*). It remains to be determined whether the heritability of adiponectin concentrations is quantile-dependent, and whether this produces significant heritability differences by sex, genotype-specific increases during rosiglitazone treatment or postprandial lipemia, and gene-environment interactions by adiposity level.

We therefore used nonparametric quantile regression (*Koenker & Hallock, 2001*; *Gould, 1992*) to test whether untransformed adiponectin concentrations exhibit quantile-dependent heritability in the narrow-sense ($h^2$) as estimated from offspring-parent ($\beta_{OP}$) and full-sib ($\beta_{FS}$) regression slopes (*Falconer & Mackay, 1996*) in a large population (Framingham Heart Study *Dawber, Meadors & Moore, 1951*; *Kannel et al., 1979*; *Splansky et al., 2007*). Untransformed concentrations were used because quantile regression does not require normality, and no biological justification has been given for its logarithmic transformation. Heritability was studied because between 5% and 9% of the variation in adiponectin is accounted for by variants within the gene encoding adiponectin (ADIPOQ) and other loci (*Dastani et al., 2012*; *Heid et al., 2010*). However, because heritability lacks the specificity of directly measured genotypes, we also examined published studies that measured genetic variants directly from the perspective of quantile-dependent expressivity to establish external validity and generalizability.

## METHODS

The methods have been described previously (*Williams, 2020c*; *Williams, 2020e*; *Williams, 2020a*; *Williams, 2020g*), but are repeated here for completeness. The data were obtained from the National Institutes of Health FRAMCOHORT, GEN3, FRAMOFFSPRING Research Materials obtained from the National Heart Lung and Blood Institute (NHLBI) Biologic Specimen and Data Repository Information Coordinating Center. The Original Framingham cohort consisted of men and women between the ages of 30 and 62 from the town of Framingham, Massachusetts (*Dawber, Meadors & Moore, 1951*). The Offspring (generation 2) Cohort consisted of 5,124 adult children of the original participants and their spouses who were first examined between 1971 and 1975, re-examined eight years later, and then every three to four years thereafter (*Kannel et al., 1979*). Children of the Offspring Cohort were recruited to form the Third Generation Cohort (*Splansky et al., 2007*). Subjects were at least 16 years of age and not self-identified as nonwhite or Hispanic. Adiponectin concentrations were measured on stored blood samples frozen at $-80\,°C$ from examination 7 of the Framingham Offspring Cohort and examination 1 of the Framingham Third Generation Cohort by ELISA (R&D Systems) with an average interassay coefficients of variation $< 5\%$ (*Zachariah et al., 2017*). The statistical analyses were approved by Lawrence Berkeley National Laboratory Human Subjects Committee (HSC) for protocol

"Gene-environment interaction vs. quantile-dependent penetrance of established SNPs (107H021)" LBNL holds Office of Human Research Protections Federal wide Assurance number FWA 00006253. Approval number: 107H021-13MR20. The original surveys were conducted under the direction of the Framingham Heart Study human use committee guidelines, with signed informed consent from all participants or parent and/or legal guardian if < 18 years of age.

## Statistics

Age and sex adjustment was performed separately for each examination of the Offspring and Third Generation Cohorts using standard least-squares regression with the following independent variables: female (0,1), age, $age^2$, female x age, and female $\times age^2$. Individual subject values were taken as the average of the residuals over all available examinations. Offspring-parent correlations and regression slopes were computed by assigning a weight of one-half to the child-father and one-half to the child-mother pair (if both parents available), and assigning a weight of one to the child-parent pair if only one parent was available. Offspring-midparental correlations and regression slopes were computed by comparing each child's age and sex-adjusted value to the average of the age and sex-adjusted parental values in those families having both parents. Full-sibling correlations were obtained by constructing all possible pairs using double entry (*Karlin, Cameron & Williams, 1981*). Unadjusted quantile regression analysis means an unadjusted dependent variable (e.g., offspring, sib) was compared to the age and sex-adjusted independent variables (i.e., parent, other sibs). The number of degrees of freedom for the standard error was adjusted to $\Sigma k_i$-2 for offspring-parent and midparental regression slopes and correlations, and $\Sigma (k_i$-1) for sibship correlations and regression slopes, where $k_i$ is the number of offspring in family $i$ and the summation is taken over all $i$, $i = 1,\ldots$, N nuclear families (*Karlin, Cameron & Williams, 1981*). Slopes are presented $\pm$SE.

Simultaneous quantile regression is a well-developed statistical procedure (*Koenker & Hallock, 2001*) that estimates the regression coefficients for multiple quantiles using linear programming to minimize the sum of asymmetrically weighted absolute residuals, and bootstrap resampling to estimate their corresponding variances and covariances (*Gould, 1992*). Simultaneous quantile regression was performed using the "sqreg" command of Stata (version. 11, StataCorp, College Station, TX) with one thousand bootstrap samples drawn to estimate the variance–covariance matrix for the 91 quantile regression coefficients between the 5th and 95th percentiles, and the post-estimation procedures (test and lincom) to test linear combinations of the slopes after estimation with $\Sigma k_i$-2 degrees of freedom for offspring-parent regression slopes and $\Sigma (k_i$-1) degrees of freedom for sibship regression slopes. Quantile-specific expressivity was assessed by: (1) estimating quantile-specific $\beta$-coefficient for the 5th, 6th,..., 95th percentiles of the sample distribution using simultaneous quantile regression (Fig. 1, the <5th and >95th percentiles ignored because they were thought to be less stable); (2) plotting the quantile-specific $\beta$ coefficients vs. the percentile of the trait distribution; and (3) testing whether the resulting graph is constant, or changes as a linear, quadratic, or cubic function of the percentile of the trait distribution using orthogonal polynomials (*Winer, Brown & Michels, 1991*). Heritability

in the narrow sense ($h^2$) was estimated as $h^2 = 2\beta_{OP}/(1+r_{spouse})$ from offspring-parent regression slopes ($\beta_{OP}$), $h^2 = \beta_{OM}$ from the offspring midparental slope ($\beta_{OM}$), and $h^2 = \{(1+8r_{spouse}\ \beta_{FS})^{0.5}-1\}/2r_{spouse}$ from full-sibs regression slopes ($\beta_{FS}$) where $r_{spouse}$ is the spouse correlation (*Falconer & Mackay, 1996*) "Quantile-specific heritability" refers to the heritability statistic ($h^2$), whereas "quantile-dependent expressivity" is the biological phenomenon of the trait expression being quantile-dependent.

When $\beta_{OP}$ for male and female offspring are included on the same graph, their quantile-specific functions compares their heritabilities at the corresponding percentiles of their separate distribution (e.g., the slope at the 50th percentile of the daughters' distribution vs. the slope at the 50th percentile of the sons' distribution). However, the adiponectin concentration at the 50th percentile of the daughters' distribution will be greater then the 50th percentile of the sons' distribution. Quantile-specific expressivity postulates that the genetic effects depend upon the adiponectin concentration. Therefore, additional displays were created using Q-Q plots (*Wilk & Gnanadesikan, 1968*) to re-plot the sons' and daughters' heritability at the same adiponectin concentrations.

In the discussion, the results from other studies were re-interpreted from the perspective of quantile-dependent expressivity using the genotype-specific mean adiponectin concentrations presented in the original articles or by extracting these values from graphs using the Microsoft Powerpoint formatting palette (version 12.3.6 for Macintosh computers, Microsoft corporation, Redmond WA) as previously described (*Williams, 2020b*).

### Data availability

The data are not being published in accordance with the data use agreement between the NIH National Heart Lung, and Blood Institute and Lawrence Berkeley National Laboratory. However, the data that support the findings of this study are available from NIH National Heart Lung, and Blood Institute Biologic Specimen and Data Repository Information Coordinating Center directly through the website https://biolincc.nhlbi.nih.gov/my/submitted/request/ (*National Heart, Lung, and Blood Institute, 2020b*). Restrictions apply to the availability of these data, which were used under license for this study. Those wishing a copy of the data set should contact the Blood Institute Biologic Specimen and Data Repository Information Coordinating Center at the above website, where they can find information on human use approval and data use agreement requiring signature by an official with signing authority for their institute. The public summary-level phenotype data may be browsed at the dbGaP study home page (*Genotypes and Phenotypes (dbGaP), 2020a*).

## RESULTS

### Traditional estimates of familial concordance and heritability

The sample characteristics displayed in Table 1 show average adiponectin were significantly higher in women than men. BMI was negatively correlated with adiponectin concentrations ($r = -0.31$) when age and sex adjusted. Spouse correlation for adjusted adiponectin concentrations was weak ($r_{spouse} = 0.04$). The offspring-parent regression slope for adjusted

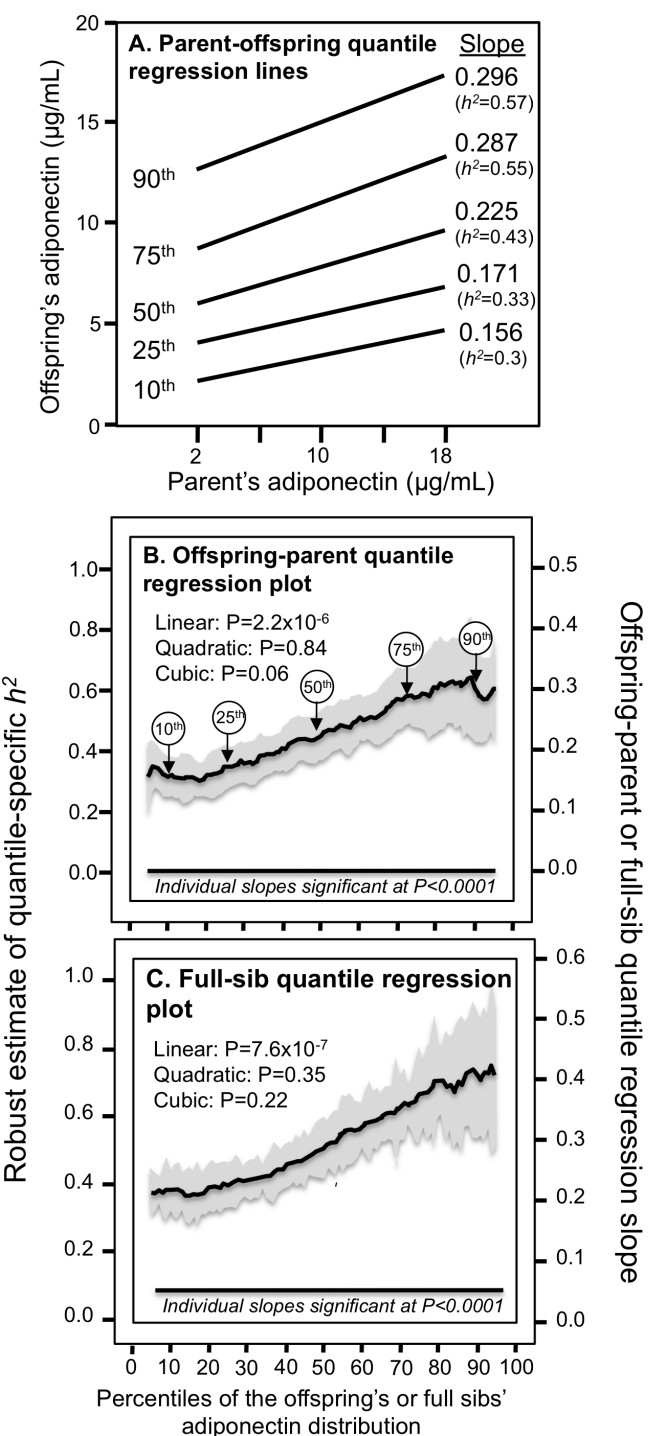

**Figure 1  Offspring-parent and full-sib quantile regression slopes.** (A) Offspring-parent regression slopes ($\beta_{OP}$) for selected quantiles of the offspring's adiponectin concentrations from 4,182 offspring-parent pairs, with corresponding estimates of heritability ($h^2 = 2\beta_{OP}/(1 + r_{spouse})$), where the correlation between spouses was $r_{spouse} = 0.04$. The slopes became (continued on next page...)

**Figure 1 (...continued)**
progressively greater (i.e., steeper) with increasing quantiles of the adiponectin distribution. (B) The selected quantile-specific regression slopes were included with those of other quantiles to create the quantile-specific heritability function in the lower panel. Significance of the linear, quadratic and cubic trends and the 95% confidence intervals (shaded region) determined by 1,000 bootstrap samples. (C) Quantile-specific full-sib regression slopes ($\beta_{FS}$) from 4,587 siblings in 1,662 sibships, with corresponding estimates of heritability as estimated by $h^2 = (8r_{spouse}\beta_{FS} + 1)^{0.5} - 1/(2r_{spouse})$. 95% confidence intervals (shaded region) determined by 1,000 bootstrap samples.

**Table 1  Sample characteristics.**

|  | Males | | Females | |
| --- | --- | --- | --- | --- |
|  | Offspring cohort | Third generation cohort | Offspring cohort | Third generation cohort |
| Age, years | 61.21 (9.63) | 40.44 (8.62) | 60.93 (9.41) | 39.91 (8.73) |
| BMI, kg/m² | 28.62 (4.62) | 27.99 (4.67) | 27.43 (5.80) | 26.03 (6.11) |
| Adiponectin, µg/mL | 7.45 (6.63) | 6.09 (3.82) | 12.59 (6.71) | 10.97 (5.77) |

adiponectin concentrations ($\beta_{OP} \pm$ SE: 0.22 ± 0.01), computed from 1718 offspring with one parent and 1232 offspring with two parents, corresponds to a heritability ($h^2$) of 0.43 ± 0.03, the same as when estimated from $\beta_{OM}$ ($\beta_{OM}$=0.43 ± 0.03). There were 4587 full-sibs in 1662 sibships with age and sex-adjusted adiponectin concentrations, whose full-sib regression slope ($\beta_{FS}$) was 0.29 ± 0.02, which from Falconer's formula, corresponds to a heritability of $h^2$=0.57 ± 0.04.

## Quantile-dependent expressivity

The $\beta_{OP}$'s (offspring-parent regression slopes) at the 10th, 25th, 50th, 75th, and 90th percentiles of the offspring's adiponectin distribution are presented in Fig. 1A, along with their corresponding heritability estimates ($h^2$= 2*$\beta_{OP}$/(1+r_{spouse})). The slopes get progressively greater with increasing percentiles of the adiponectin distribution. The heritability at the 90th percentile was 0.57, which is 89.6% greater than the heritability at the 10th percentile ($P_{difference} = 0.001$). The quantile-specific heritability plot of Fig. 1B presents these slopes, along with those of the other percentiles between the 5th and 95th percentiles. They show heritability increased linearly (i.e., slope ± SE: 0.0038 ± 0.0008, $P_{linear}$=2.2 ×10⁻⁶) with increasing percentiles of the offspring's distribution. There was no significant evidence of nonlinearity (i.e., $P_{quadratic} = 0.84$; $P_{cubic} = 0.06$). Quantile-specific heritability was individually significant ($P \leq 7.2 \times 10^{-7}$) for all percentiles of the offspring's distribution. If the heritability over all quantiles was constant, then the line segments would all be parallel in Figs. 1A, and 1B would show a flat line having zero slope. Figure 1C displays the quantile regression analysis for $h^2$ estimated from full-sib regression slopes ($\beta_{FS}$). Each one-percent increase in the adiponectin distribution was associated with a 0.0052 ± 0.001 increase in heritability and a 0.0026 ± 0.0005 increase in the full-sib regression slope ($P_{linear}$=7.6 ×10⁻⁷).

Significant quantile-dependent expressivity was replicated when 506 sibships from the Offspring Cohort and 1156 sibships from the Third Generation Cohorts were analyzed

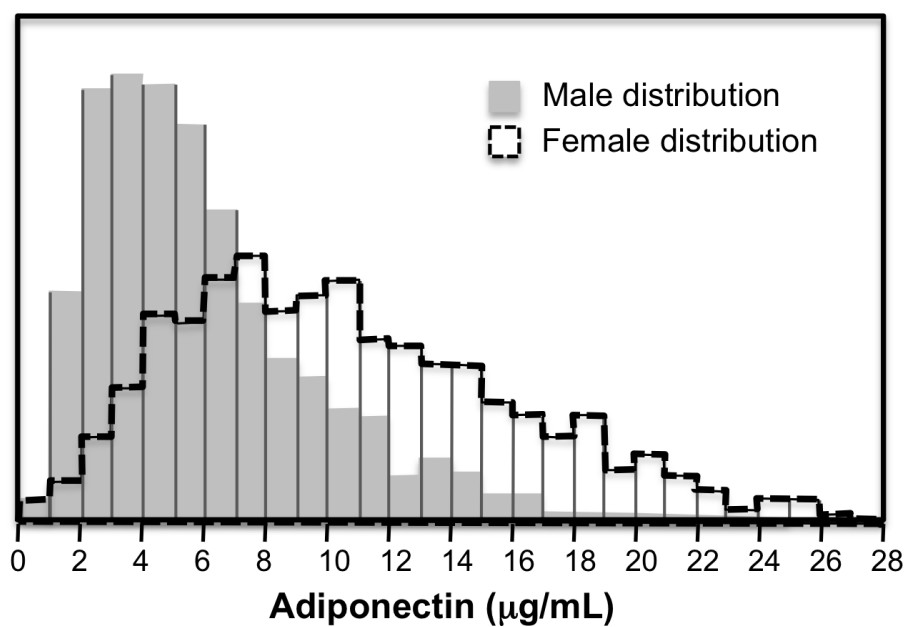

**Figure 2** **Distribution of fasting adiponectin concentrations in males and females.**

separately, i.e., $\beta_{FS}$ increased $0.0023 \pm 0.0011$ in the Offspring Cohort ($P = 0.04$) and $0.0028 \pm 0.0006$ in the Third Generation Cohort ($P = 8.0 \times 10^{-6}$) for each one-percent increment in the sibs' adjusted adiponectin concentrations.

### Male–female differences in heritability

The preceding analyses showed that adiponectin heritability increased with increasing percentiles of the offspring distribution for the combined sample of male and female age- and sex-adjusted offspring. Figure 2 however, shows that the female adiponectin distribution is shifted towards to the right of the males. Correspondingly, the analyses of Fig. 1B suggest that female heritability should be greater than that of the males. In fact, heritability as classically estimated by standard regression was higher in females than males for adiponectin ($0.53 \pm 0.05$ vs. $0.33 \pm 0.03$, $P < 10^{-15}$) and Fig. 3A shows that the quantile-specific heritability was higher in females than males at each percentile of their respective distribution. Adiponectin heritability was significantly greater in females than males ($P < 0.05$) for each percentile between the 8th and the 77th percentile.

From the perspective of quantile-dependent expressivity, the problem with Fig. 3A is that comparing male and female heritability at their 10th percentiles means comparing the male heritability at an unadjusted adiponectin concentration of 2.25 μg/ml with the female heritability at an unadjusted concentration 4.25 μg/ml, comparing their heritability at their 50th percentile means comparing the male heritability at 5.18 μg/ml with the female heritability at 9.98 μg/ml, and comparing their heritability at the 90th percentiles means comparing the male heritability at 11.41 μg/ml with the female heritability at 18.91 μg/ml. Specifically, quantile-dependent expressivity predicts an increase in heritability with increasing adiponectin concentrations. Therefore the male and female heritability

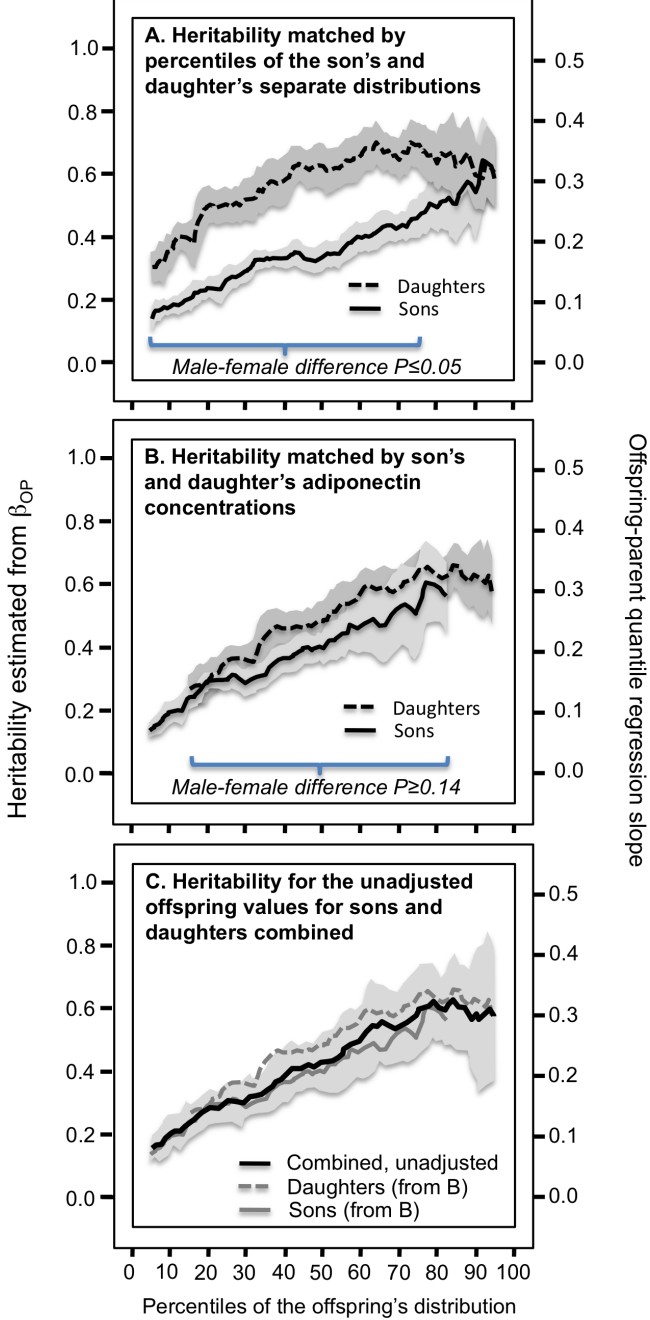

**Figure 3** **Offspring-parent quantile regression slopes ($\beta_{OP}$) in male and female offspring separately.**
(A) Offspring-parent regression slopes ($\beta_{OP}$) in male and female offspring separately from age- and sex-adjusted parent-son and parent-daughter pairs, showing their significant difference when the slopes are compared at their corresponding percentiles (the sons' vs. the daughters' $\beta_{OP}$ compared at the 5th percentile of separate distributions, the 6th percentile of their separate distributions, ..., 95th percentile of their separate distributions). Shaded area designates ±SE; (continued on next page...)

**Figure 3 (…continued)**
(B) Offspring-parent regression slopes ($\beta_{OP}$) in male and female offspring showing the significant difference is eliminated when compared at their corresponding adiponectin concentrations (the sons' vs. the daughters' $\beta_{OP}$ translated using quantile-quantile (Q-Q) plots to the adiponectin concentrations at the 5th percentile of their combined distribution, the 6th percentile of their combined distribution, …, 95th percentile of their combined distribution). Shaded area designates ±SE. (C) Offspring-parent regression slopes for sons and daughters combined without adjustment for sex, showing the unadjusted analysis provides a simpler description of the quantile increase based solely on the percentiles of their unadjusted adiponectin concentrations. Note that the separate curves for sons' and daughters' fall fully within the 95% confidence interval (shaded area) for their combined sex-unadjusted analysis.

graphs were re-plotted to the same adiponectin concentrations in Fig. 3B using quantile–quantile (Q-Q) plots (see methods). This eliminated the significant differences between the male and female heritability plots. Similarly, Figs. 4A and 4B present the analyses for the full-sib estimates of heritability showing substantial differences between the male and female graphs when matched by the percentiles of their corresponding age and sex-adjusted distribution that are eliminated when matched by their corresponding unadjusted adiponectin concentrations. Figs. 3C and 4C show that a simple plot of the unadjusted quantile regression slopes by percentiles of the offspring or sib distribution includes the re-plotted male and females graphs of Figs. 3B and 4B within its 95% confidence interval.

# DISCUSSION

Our analyses suggest that plasma adiponectin concentrations exhibit quantile-dependent expressivity. The finding was replicated using the full-sib regression analyses in the Framingham Offspring Cohort ($P_{linear} = 0.04$) and the Framingham Third Generation Cohort separately ($P_{linear} = 8.0 \times 10^{-6}$). Moreover, the stronger adiponectin heritability in female than male offspring can be largely attributed to quantile-dependent expressivity and the females' higher concentrations (Figs. 3 and 4). A similar analytic approach was previously used to show that quantile-dependent expressivity explained the larger male than female postprandial triglyceride difference for the *APOA5* −1131 T>C polymorphism (*Williams, 2020b*). These examples suggest pro forma statistical adjustment for sex may conceal important properties of a trait's heritability. In fact, the replotted heritability of Figs. 3C and 4C show the unadjusted offspring adiponectin concentrations provided the simplest representation of their quantile-specific heritabilities.

Women have higher adiponectin concentrations due at least in part to the adiponection-lowering effects of testosterone (*Berra et al., 2006*). Whereas sex-differences in adiponectin concentrations are consistently reported (*Liu et al., 2008*; *Hicks et al., 2007*; *Comuzzie et al., 2001*; *Vaughan et al., 2015*; *Chuang et al., 2004*; *Dosaev, Prakash & Livshits, 2014*; *Henneman et al., 2010*; *Guo et al., 2006*; *Ling, Waterworth et al., 2009*; *Vasseur et al., 2002*; *Cesari et al., 2007*), sex-differences in their heritabilities are not. This we attribute to their reliance on statistical procedures that require normally distributed data and logarithmic or other data transformations. These transformations accentuate the slope at lower phenotype values and diminish the slope at higher values. For example, using the Framingham data reported here, the traditional (nonquantile) offspring-parent slope ($\beta_{OP}$ ±SE) for female

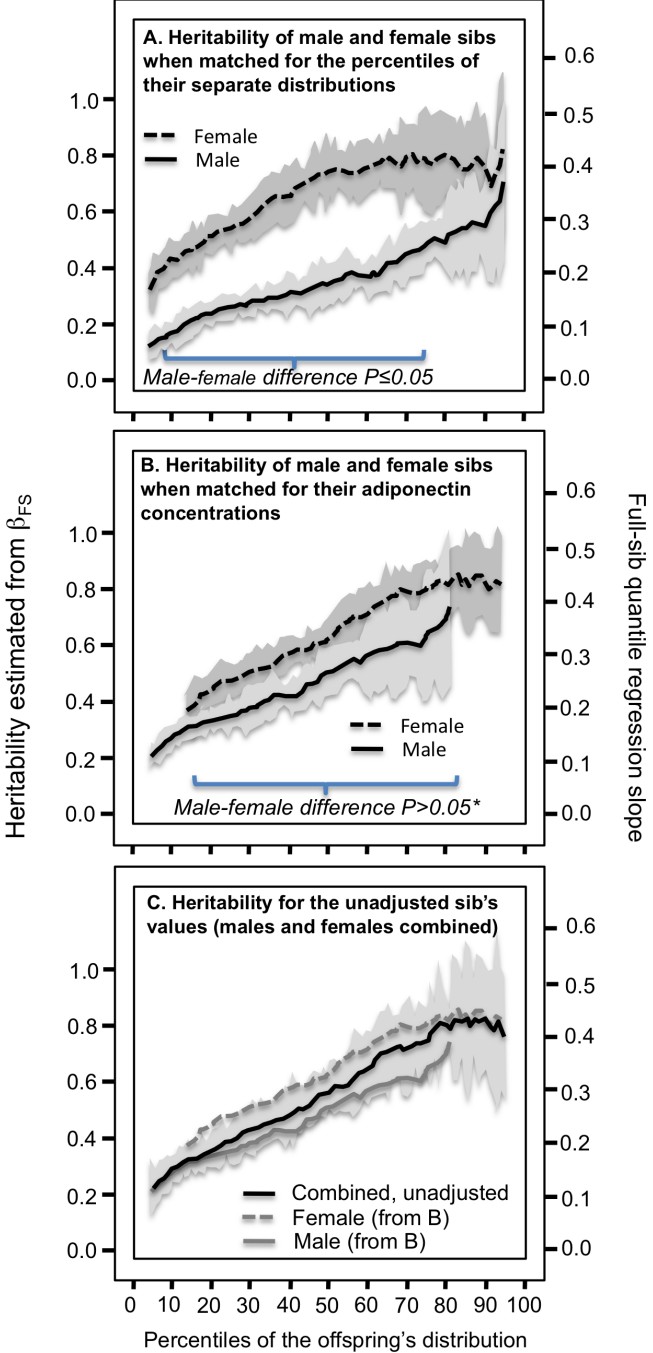

**Figure 4   Full-sib quantile regression slopes ($\beta_{FS}$) in male and female offspring separately.** (A) Analyses showing that the full sib regression slopes ($\beta_{FS}$) was greater in female than male siblings when matched by their corresponding percentiles, (B) but not when matched by their corresponding adiponectin concentrations, and (C) that a simpler graph of their combined male and female sibs, unadjusted for sex, includes their separate curves within its 95% confidence interval. See legend to Fig. 3 for details. *exceptions were $P = 0.05$ at the 39th, $P = 0.04$ at the 40th, and $P = 0.03$ at the 42nd percentiles.

vs. male offspring was $0.2733 \pm 0.0238$ vs. $0.1697 \pm 0.0171$ ($P_{\text{difference}}<10^{-15}$) for the untransformed data and $0.3221 \pm 0.0248$ vs. $0.3255 \pm 0.0294$ for the log-transformed data ($P_{\text{difference}} = 0.93$). The important point is that quantile regression and its bootstrap-derived standard errors do not require a normal distribution (*Koenker & Hallock, 2001*; *Gould, 1992*). There is no biological imperative to logarithmically or otherwise transform the data. That is not to say that quantile-regression is invariant to data transformations, which they are not (Fig. S1), but rather the rationale for transformations should ideally be biologically based, not statistically based, and its consequences acknowledged.

All the major genomewide association studies were performed on logarithmic (*Richards et al., 2009*; *Jee et al., 2010*; *Ling et al., 2009*; *Gu, 2009*) or z-score transformed adiponectin concentrations (*Bouatia-Naji et al., 2006*). Our results suggest this statistical accommodation may work against the goal of identifying SNPs affecting adiponectin concentrations. Specifically, Fig. 1 suggests that the transformation accentuates the genetic effect at low concentrations (where the genetic effects are weakest) and diminishes the genetic effect at higher values concentrations (where the genetic effects are strongest). Our previous analyses (*Williams, 2012*; *Williams, 2020c*; *Williams, 2020e*; *Williams, 2020a*; *Williams, 2020b*) suggest this concern is also apropos to lipoproteins and adiposity GWAS.

Important caveats to our analysis of phenotypes in family sets are: (1) heritability lacks the specificity of directly measured genotypes even if it is a more inclusive measure of genetic effects; and (2) Falconer's formula probably do not adequately address the true complexity of the genetics and shared environment affecting adiponectin concentrations. These concerns can be partly addressed by re-analyzing published studies that measured genetic variants directly from the perspective of quantile-dependent expressivity. They include multiple examples where the paper's original interpretation from the perspective of precision medicine or gene-environment interactions might be more simply explained by a single underlying phenomenon: quantile-dependent expressivity. Results are presented in their reported units.

## Pharmacogenetics

There is an important distinction between quantile-dependence and pharmacogenetics. Pharmacogenetics attempts to use genetic markers that identify patients most likely to benefit from specific treatments to individualize drug prescriptions. Quantile-dependent expressivity postulates that drugs alter the phenotype (e.g., increase adiponectin concentrations), which in turn alters the expressivity of genetic variants. More simply stated, genetic markers merely track the increase in heritability with increasing adiponectin concentrations.

For example, rosiglitazone is a thiazolidinedione derivate that increases serum adiponectin concentration by increasing adiponectin transcription (*Kang et al., 2005*). *Kang et al. (2005)* reported significantly smaller increases in adiponectin concentrations in GG homozygotes of the at position 45 (rs2241766) of the ADIPOQ gene than carriers of the T allele after 166 T2DM's received 4 mg/day of rosiglitazone for 12 weeks ($P < 0.003$, Fig. 5A histogram). Heterozygotes had an intermediate response. Alternatively, from the perspective of quantile-dependent expressivity (Fig. 5A line graph) there were substantially

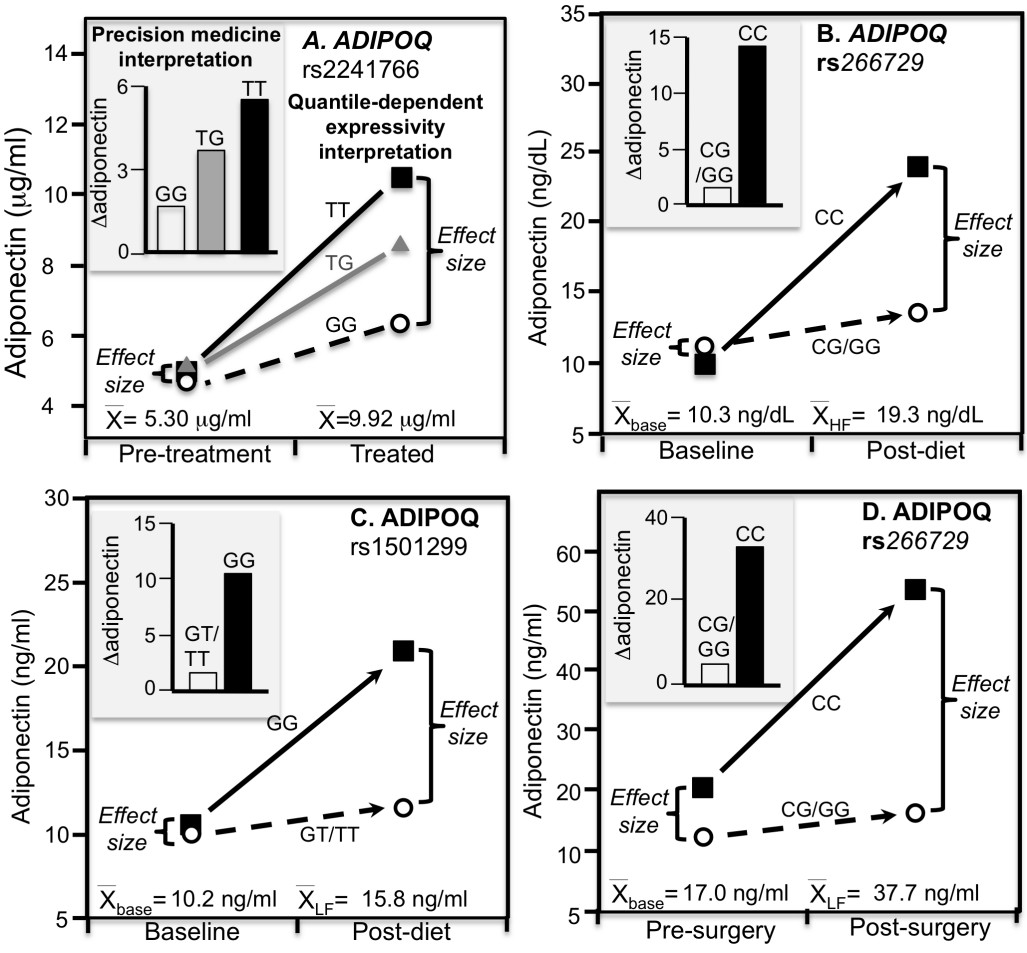

**Figure 5 Precision medicine perspective of *ADIPOQ* genotype-specific adiponectin differences (histogram inserts) vs. quantile-dependent expressivity perspective (line graphs).** Precision medicine perspective of *ADIPOQ* genotype-specific adiponectin differences (histogram inserts) vs. quantile-dependent expressivity perspective (line graphs showing larger genetic effect size when average adiponectin concentrations were high) for: (A) Kang et al.'s 2005 report (*Kang et al., 2005*) on the effect of 12-weeks 4 mg/-day of rosiglitazone treatment in 25 GG rs2241766 homozygotes and 86 T-allele carriers with T2DM; (B) de Luis et al.'s 2020 report (*De Luis et al., 2020*) on the pooled effect of switching from a basal to a 27%- or 38%-fat hypocaloric diet in 169 rs266729 CC homozygotes and 114 G-allele carriers; (C) Aller et al.'s 2019 report (*Aller et al., 2019*) on the pooled effect of switching from a basal to a standard or high-protein extreme hypocaloric diet in 122 rs1501299 GG homozygotes than 147 T-allele carriers; (D) De Luis et al's 2018 report (*De Luis et al., 2018*) on the effect of 41.9 kg weight loss from biliopancreatic diversion surgery in 84 rs266729 CC homozygote and 65 G-allele carriers who were morbidly obesity. The histograms and line graphs were derived from the mean concentrations presented in the original manuscripts.

greater differences in adiponectin concentrations between genotypes at the end of treatment than at baseline (TT minus GG difference: $4.12 \pm 1.30$ vs. $0.27 \pm 0.79\ \mu g/ml$) in accordance with the significantly higher mean adiponectin concentrations after treatment than before ($9.92 \pm 0.53$ vs. $5.30 \pm 0.37\ \mu g/ml$, $P < 0.001$).

## Gene-environment interactions

There are multiple reports of adiposity modulating genetic influences on adiponectin concentrations, or equivalently, that these polymorphisms modulated the effects of adiposity on adiponectin concentrations (Figs. 5B–5D and Figs. 6A–6C histograms). These include the rs266729 (−11,377C>G) polymorphism located in the proximal promoter region of the ADIPOQ gene and which functionally regulates adiponectin promoter activity and adiponectin levels (*Gu, 2009*; *Bouatia-Naji et al., 2006*), rs1501299 (+276T>G) in ADIPOQ's intron 2, and the aforementioned rs2241766 in ADIPOQ's exon 2.

*De Luis et al. (2020)* reported significantly greater increases in adiponectin concentration in CC homozygotes than G-carriers of the ADIPOQ rs266729 gene polymorphism when participants switched from a basal diet to either a 27% low-fat hypocaloric (CC vs. G-carriers: $16.1 \pm 2.8$ vs. $1.3 \pm 1.0$ ng/dL, $P = 0.03$) or a 38% high fat hypocaloric diet ($10.6 \pm 2.0$ vs. $1.8 \pm 1.0$ ng/dL, $P = 0.01$) for three months (Fig. 5B histogram, pooled across diets). Both diets produced significant weight loss: $4.5 \pm 0.9$ kg on the high-fat and $4.1 \pm 0.9$ kg on the low-fat diet. Alternatively, on the high-fat diet, the adiponectin difference between genotypes was greater after weight loss ($8.3 \pm 0.8$ ng/dL) when the overall average concentration was higher ($16.9 \pm 0.4$ ng/dL) vis-à-vis before weight loss ($-0.5 \pm 0.7$ ng/dL) when overall average concentration was lower ($9.8 \pm 0.3$ ng/dL). Similarly, on the low-fat diet there was a larger adiponectin difference between genotypes after weight loss ($14.0 \pm 1.3$ ng/dL) at the higher average concentration ($21.5 \pm 0.8$ ng/dL) vis-à-vis before weight loss ($-1.8 \pm 1.1$ ng/dL) at the lower average concentration ($10.8 \pm 0.7$ ng/dL) (*De Luis et al., 2020*), suggesting that quantile-dependent expressivity may have contributed to the genotype-specific increases (Fig. 5B line graph for the pooled results).

From the same laboratory, *Aller et al. (2019)* reported greater 9-month increases in adiponectin concentrations in GG homozygotes of the rs1501299 gene than T-allele carriers when switching from their basal diet to one of two severe hypocaloric diets: a standard version and a high-protein low-carbohydrate version. Both diets increased adiponectin significantly in GG homozygotes (standard: 10.9 ng/ml, $P < 0.05$; high-protein: 10.1 ng/ml, $P < 0.05$) but not in carriers of the T allele (standard: 0.6 ng/ml; high-protein: 2.6 ng/ml). Their pooled results are presented in Fig. 5C histogram. However, for both diets average adiponectin concentrations were higher after 9-month weight loss (standard: $15.4 \pm 0.5$ ng/ml; high-protein: $16.3 \pm 0.4$ ng/ml) than at baseline (standard: $10.3 \pm 0.5$ ng/ml; high-protein: $10.1 \pm 0.3$ ng/ml), and in accordance with quantile-dependent expressivity, the difference between GG and T-allele carriers was greater for the higher average concentrations after weight loss (standard: $11.5 \pm 1.0$ ng/ml; high-protein: $7.3 \pm 0.9$ ng/ml) than at the low average concentrations at baseline (standard: $1.2 \pm 0.9$ ng/ml; high-protein: $-0.2 \pm 0.6$ ng/ml). The line graph of Fig. 5C presents this quantile-dependent interpretation for the pooled sample.

*De Luis et al. (2018)* also reported that rs266729 CC homozygotes had significantly greater adiponectin increases than G-carriers when 149 morbidly obese patients lost an average of 41.9 kg during the three years following biliopancreatic diversion surgery (Fig. 5D histogram, $33.2 \pm 0.4$ vs. $4.7 \pm 0.2$ ng/ml; $P = 0.01$). From the perspective of

quantile-dependent expressivity, the genetic effect size between CC homozygotes and G-allele carriers increased as mean adiponectin concentration increased from $17.0 \pm 0.4$ ng/ml pre-surgery ($8.7 \pm 0.8$ ng/ml difference between genotypes), to $27.1 \pm 0.5$ ng/ml one-year post surgery ($22.5 \pm 1.0$ ng/ml genotype difference), $31.8 \pm 0.4$ ng/ml two-years post surgery ($29.8 \pm 0.9$ ng/ml genotype difference), and $37.7 \pm 0.5$ ng/ml three-years post surgery ($37.1 \pm 1.1$ ng/ml genotype difference).

A third study by *De Luis et al. (2019)* reported that rs266729 CC homozygotes had significantly greater adiponectin increases than G-carriers (Fig. 6A histogram, $10.4 \pm 3.1$ vs. $-1.3 \pm 1.0$ ng/dL, $P = 0.01$) when 83 obese patients lost an average of $3.5 \pm 0.6$ kg after a 3-month Mediterranean-type hypocaloric diet. Again, from the perspective of quantile-dependent expressivity, the genetic effect size between CC homozygotes and G-allele carriers increased as mean adiponectin increased from the pre-diet $23.8 \pm 0.6$ ng/dL average ($10.2 \pm 1.1$ ng/dL genotype difference) to the $28.5 \pm 0.4$ ng/dL post-diet average ($21.9 \pm 0.9$ ng/dL difference).

Cross-sectionally, *Divella et al. (2017)* reported that the difference in adiponectin concentration between obese and normal weight colorectal cancer patients was greater in rs266729 CC homozygotes than CG/GG genotypes ($44.5 \pm 10.4$ vs. $32.3 \pm 10.1$ ng/ml, Fig. 6B histogram). Consistent with quantile-dependent expressivity, the associated line graph shows that the difference between genotypes increased as mean adiponectin concentrations increased from $46.3 \pm 4.2$ ng/ml in obese (genotype difference $22.1 \pm 8.9$ ng/ml), $51.8 \pm 5.5$ ng/ml in overweight ($30.6 \pm 16.3$ ng/ml difference, not displayed), to $94.6 \pm 5.7$ ng/ml in normal weight patients ($34.3 \pm 11.5$ ng/ml difference).

*Garcia-Garcia et al. (2014)* concluded that adiponectin levels were modulated by the interaction between BMI and ADIPOQ −11391G/A SNP on the basis of a significant adiponectin difference between GA and GG genotypes in the 1st ($1.30 \pm 0.66$ μg/ml, $P = 0.03$) but not 2nd ($0.2 \pm 0.29$ μg/mL) nor 3rd BMI tertiles ($0.2 \pm 0.24$ μg/ml), consistent with quantile-dependent expressivity given that mean adiponectin concentrations were significantly higher in the 1st ($4.20 \pm 0.28$ μg/ml) than the 2nd ($3.09 \pm 0.15$) or 3rd BMI tertiles ($2.30 \pm 0.12$ μg/ml).

*Berthier et al. (2005)* reported that visceral adiposity modulated the effect of the rs2241766 *ADIPOQ* gene polymorphism on adiponectin concentrations. Otherwise stated, Fig. 6C histogram (estimated from their figure 1) shows the effect of visceral fat was greater in carriers of the G-allele than TT homozygotes. From the perspective of quantile-dependent expressivity, the genetic effect size was greater in the less-viscerally obese than viscerally obese subjects (6.0 vs. 0.4 μg/L) in accordance with their higher average adiponectin concentrations.

## Sex-specific genetic effects

Quantile-dependent expressivity, in conjunction with the higher average adiponectin concentrations in women than men ($6.04 \pm 0.10$ vs. $4.08 \pm 0.10$ μg/ml), might explain *Riestra et al. (2015)* report that ADIPOQ variants rs6444174, rs16861205, rs1403697, and rs7641507 were strongly associated with serum adiponectin concentrations in women but not men.
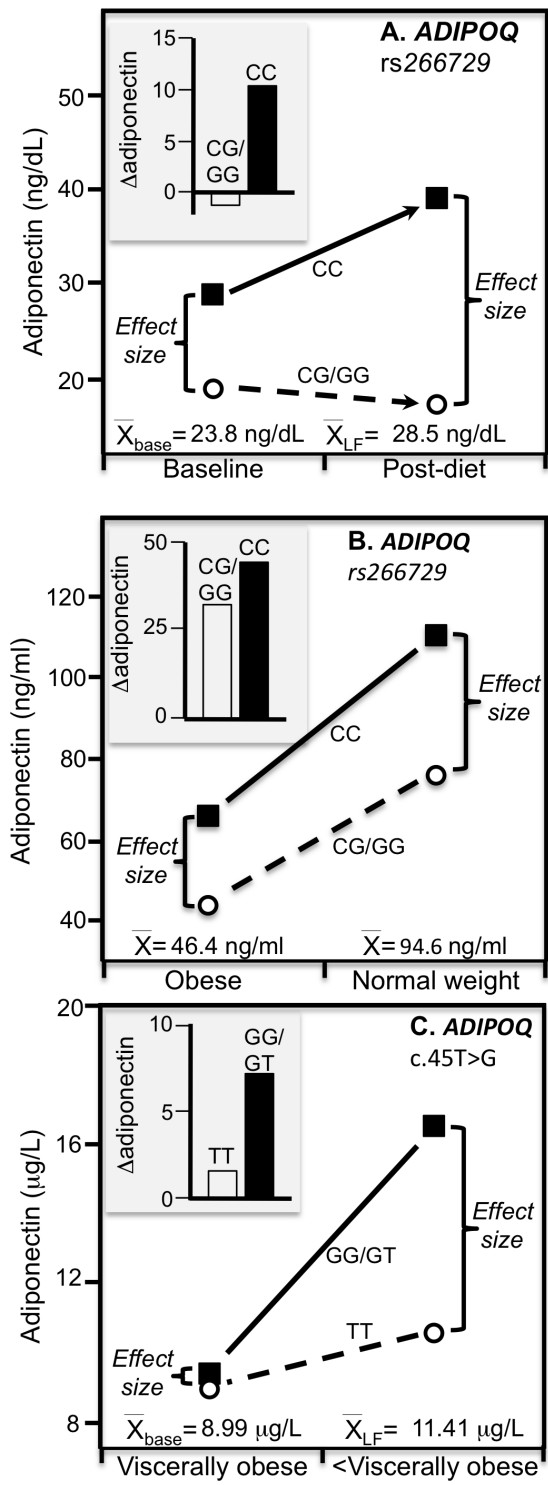

**Figure 6** **Precision medicine perspective of *ADIPOQ* genotype-specific adiponectin differences (histogram inserts) vs. quantile-dependent expressivity perspective (line graphs).** Precision medicine perspective of *ADIPOQ* genotype-specific adiponectin differences (histogram inserts) vs. quantile-dependent expressivity perspective (line graph showing larger genetic (continued on next page...)

**Figure 6 (…continued)**
effect size when average adiponectin concentrations were high) for: (A) de Luis et al. 2019 report (*De Luis et al., 2019*) on a 3-month Mediterranean-type hypocaloric diet in 48 rs266729 CC homozygotes and 45 G-allele carriers; (B) Divella et al. 2017 report (*Divella et al., 2017*) on the cross-sectional difference between being obese and nonobese in 30 rs266729 CC homozygotes and 73 G-allele carriers with colon cancer; and (C) Berthier et al. (*Berthier et al., 2005*) 2005 report of the cross-sectional difference between high and low visceral adiposity (computed tomography $\geq$ 130 vs. < 130 cm$^2$) in 26 rs2241766 TT-homozygotes vs. 117 male G-allele carriers. The histograms and line graphs were derived from the mean concentrations presented in the original manuscripts.

## Postprandial lipemia

The dependence of genetic effects on mean adiponectin concentrations has also been demonstrated within individuals during their postprandial response. Carriers of the 45TT (rs2241766) and 276GT/TT (rs1501299) *ADIPOQ* haplotype have a higher T2DM and cardiovascular disease risk than noncarriers. As derived from Musso et al.'s report (*Musso et al., 2008*), Fig. 7 shows that the haplotype's blunted affects on the postprandial adiponectin concentrations following an oral fat load were linearly related to the average adiponection concentrations at time t (linear regression, 4 df, $P = 0.0002$).

## Caveats and limitations

None of the SNPs identified to date explain any more than a few percent of adiponectin heritability, which means that the effects of any particular SNP is not necessarily constrained by the results of Fig. 1. Exceptions to Fig. 1 include *Hara et al. (2002)* report of significant adiponectin differences between *ADIPOQ* rs1501299 genotypes for obese Japanese whose mean concentrations were low, but not lean Japanese whose mean adiponectin concentrations were higher; and Gupta et al. reported that the *ADIPOQ* rs2241766 polymorphism significantly affected adiponectin in patients with nonalcoholic fatty liver disease but not controls despite the lower mean concentration of the patients (4.8 vs. 7.2 µg/ml) (*Gupta et al., 2012*). We also acknowledge that the simple estimates of $h^2$ from Falconer's formula probably do not adequately describe adiponectin inheritance (*Falconer & Mackay, 1996*), i.e., those derived from $\beta_{OP}$ may include shared environmental effects, and those derived from $\beta_{FS}$ may include shared environment and dominance effects and unmet restrictions on assortative mating. We note that the analyses were based on total rather than the biologically more active high molecular weight adiponectin. Finally, quantile-dependent expressivity represents an alternative interpretation to the gene-environment and precision medicine interpretations presented by others, but our analyses do not negate the original interpretation. Our analyses do not address the relationships of adiponectin to disease risk factors or endpoints, and therefore cannot provide insight to adiponectin paradox regarding all-cause and cardiovascular mortality (*Menzaghi & Trischitta, 2018*).

   In conclusion, heritability of adiponectin concentrations is quantile-dependent, which appears to explain the stronger heritability in women in accordance with their higher concentrations, and is consistent with the interactions of genes with thiazolidinedione, adiposity, and postprandial changes reported by others. Prior reports of adiponectin heritability overlooked the effects of sex on heritability because of their use parametric

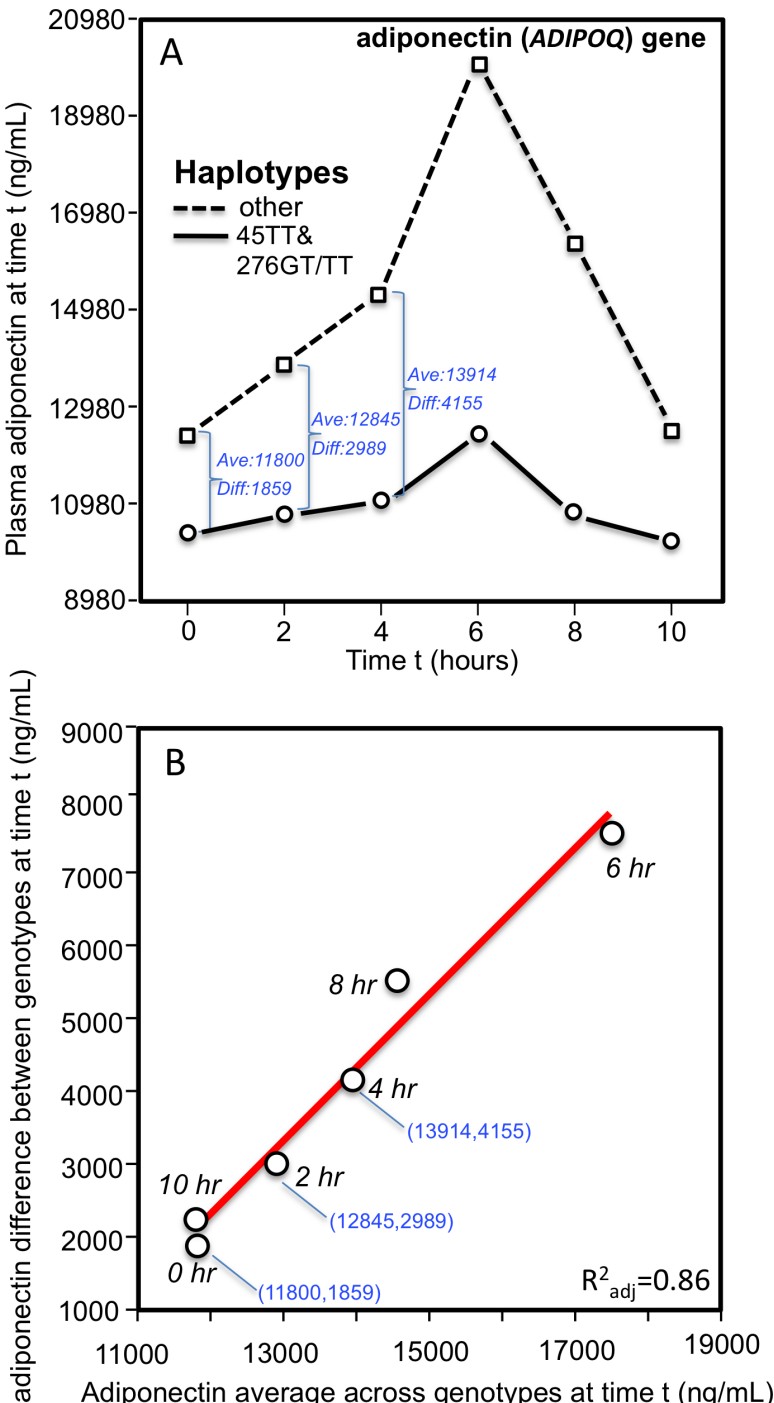

**Figure 7** **Adiponectin response to an oral fat tolerance test by 45TT (rs2241766) and 276GT/TT (rs1501299) *ADIPOQ* haplotypes.** (A) Re-rendering of *Musso et al.*'s *2008* published adiponectin response to an oral fat tolerance test by 45TT (rs2241766) and 276GT/TT (rs1501299) *ADIPOQ* haplotypes (B) regression plot showing the genotypes difference (dependent variable) increased linearly with increasing adiponectin concentrations (independent variable). The genotype-specific mean concentrations were extracted from their figure 1C using the Microsoft PowerPoint formatting palette.

statistics requiring logarithmic transformations. Genome-wide association studies of adiponectin also exclusively report on logarithmically transformed concentrations. Should we have chosen to log-transform adiponectin concentrations, the analyses would still have shown quantile-specific effects, but with heritability decreasing with increasing concentrations (Fig. S1). We analyzed untransformed adiponectin concentrations because quantile-regression does not require normality, and no biological rationale has been proposed for their logarithmic transformation. Parenthetically, the significant interactions reported by *Kang et al. (2005)*, *De Luis et al. (2020)*, *De Luis et al. (2018)*, *De Luis et al. (2019)*, *Divella et al. (2017)*, *Aller et al. (2019)* and *Garcia-Garcia et al. (2014)* were all based on untransformed adiponectin concentrations.

**Abbreviations**

| | |
|---|---|
| *ADIPOQ* | Adiponectin, C1Q And Collagen Domain Containing |
| *APOA5* | Apolipoprotein A5 gene |
| $\beta_{FS}$ | Full-sib regression slope |
| $\beta_{OM}$ | Offspring mid-parental regression slope |
| $\beta_{OP}$ | Offspring-parent regression slope |
| **BMI** | Body mass index |
| **ELISA** | Enzyme-linked immunosorbent assay |
| **GWAS** | Genome-wide association studies |
| $h^2$ | Heritability in the narrow sense |
| **NHLBI** | National Heart Lung and Blood Institute |
| **Q-Q** | plot Quantile-quantile plot |
| **SD** | Standard deviation |
| **SE** | Standard error |
| **SNP** | Single nucleotide polymorphism |
| **T2DM** | Type 2 diabetes mellitus |

### Funding

This research was supported by NIH grant R21ES020700 from the National Institute of Environmental Health Sciences, and an unrestricted gift from HOKA ONE ONE. The funders had no role in study design, data collection and analysis, decision to publish, or preparation of the manuscript.

### Grant Disclosures

The following grant information was disclosed by the author:
NIH: R21ES020700.
HOKA ONE ONE.

### Competing Interests

The authors declare there are no competing interests.

## Author Contributions

- Paul T. Williams conceived and designed the experiments, performed the experiments, analyzed the data, prepared figures and/or tables, authored or reviewed drafts of the paper, and approved the final draft.

## Human Ethics

The following information was supplied relating to ethical approvals (i.e., approving body and any reference numbers):

These analyses were approved by Lawrence Berkeley National Laboratory Human Subjects Committee (HSC) for protocol "Gene-environment interaction vs. quantile-dependent penetrance of established SNPs (107H021)" (Approval number: 107H021-13MR20). LBNL holds Office of Human Research Protections Federal wide Assurance number FWA 00006253.

## Data Availability

The data are not being published in accordance with the data use agreement between the NIH National Heart Lung, and Blood Institute and Lawrence Berkeley National Laboratory. However, the data that support the findings of this study are available from NIH National Heart Lung, and Blood Institute Biologic Specimen and Data Repository Information Coordinating Center directly through the website https://biolincc.nhlbi.nih.gov/my/submitted/request/. (*National Heart, Lung, and Blood Institute, 2020b*) Restrictions apply to the availability of these data, which were used under license for this study.

For access to the data, please contact the Blood Institute Biologic Specimen and Data Repository Information Coordinating Center to find information on the human use approval and data use agreement requiring signature by an official with signing authority for their institute. The public summary-level phenotype data may be browsed at the dbGaP study home page (*Genotypes and Phenotypes, 2020a*).

## Supplemental Information

Supplemental information for this article can be found online at http://dx.doi.org/10.7717/peerj.10099#supplemental-information.

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
