# Peer review of "Quantile-dependent expressivity of plasma adiponectin concentrations may explain its sex-specific heritability, gene-environment interactions, and genotype-specific response to postprandial lipemia"

_PeerJ, doi:10.7717/peerj.10099_

## Round 0.1 · original submission · Minor Revisions

Sorry for the delay in processing this article; I think the academic community is overloaded with reviewing work in these COVID-19 times. As you will see, there are two distinct views here: one clear acceptance (with a few suggestions) and one rather more cautious opinion. My own view is that the second is actually not really a 'major' revision, but rather a 'clarification and commentary' issue. Hence, I would suggest that you consider the comments of both, and revise your manuscript accordingly.

Extra analysis/work is not required.

I enjoyed reading this, and provided your rebuttal is clear and detailed, I do not imagine I will be re-sending this for review.

Reviewer 1 ·

Basic reporting

The author has used clear and appropriate language throughout the manuscript. He has done a commendable job in communicating quite a complex concept "quantile-dependent expressivity" and provided many examples to illustrate how it influences plasma adiponectin concentrations. The available literature is sufficiently described and cited, and there are very few typographical errors in the manuscript. The manuscript is appropriately structured, and figures and tables are clear and illustrate the results in sufficient detail. Raw data has been shared, and the results are relevant to the hypotheses being tested.
In the introduction, could the author briefly speculate as to the factors that might have contributed to such a large range of heritability estimates for adiponectin concentrations?

Experimental design

The research question is well-defined, and the measurements and statistical analyses have been both described and performed satisfactorily. This paper is quite unique in that the statistical conclusion reached in the first dataset (Framingham) is subsequently applied to several examples in the literature (eg Kang et al, DeLuis et al) to illustrate the concept.
Overall the research is robust and very well-performed, and this has been one of the best manuscripts I have reviewed in some time.

Validity of the findings

As stated above, in addition to its robust methodology, the major strength of this manuscript was application of the concept to available examples in the literature.
I wondered aloud if there were any genetic studies published in which there is no evidence for "quartile-dependent expressivity". To his credit, the author has provided some examples in lines 431-436. I was also pleased to read about the potential for log-transformation of adiponectin concentrations to negatively influence GWAS results.
The conclusions are well-stated, but I was left wondering about the molecular machinery underlying quartile-dependent expressivity. Can the author elaborate (even briefly) on how a sex-dependent factor (such as testosterone) can not only reduce adiponectin concentrations AND reduce heritability? For example, is this likely due to occupation of an element upstream of ADIPOQ by the ligand-bound androgen receptor, which may subsequently suppress any other potential influences (like diet or inflammation) on adiponectin gene transcription? Are there prior examples in the literature, and has the molecular basis for these been described in any detail?

Additional comments

Your imaginative manuscript was presented to a very high standard, and as a researcher who has studied adiponectin polymorphisms and their influence on adiponectin concentrations, I really enjoyed reading about the concept of quartile-dependent expressivity and seeing it being applied to examples in the literature. I'm very happy to recommend acceptance for your manuscript.

Reviewer 2 ·

Basic reporting

There are several typos all over the manuscript, but in particular in the introduction.
To give to the reader a complete picture of adiponectin and its role in health and disease, the authors should not forget to cite the “adiponectin paradox”, describing the role of adiponectin on cardiovascular events and mortality (Menzaghi et al., Diabetes 2018). This is important in the light of heritability and its consequence. Could heritability of quantile-dependent expressivity of adiponectin be a possible explanation for the adiponectin paradox?

Experimental design

In the results section there is no mention at all for figures 5, 6 and 7. It is not really clear how the authors have revised several papers (Ref. 28-37) according to the “quantile-dependent expressivity” of adiponectin. Have they get raw data from the above cited papers’ authors in order to analyze data and produce figures? All this part needs to be accurately explained.

Validity of the findings

Most part of the discussion is a revision of previously published papers and are not really supported by the original results (see the above mentioned concern).

Additional comments

This report on the quantile-dipendent expressivity of adiponectin concentration could be interesting regarding the approach of using quantile expression of a trait and the results obtained are sound for sex specific heritability. Completely speculative are data on gene-envinromental interactions and genotype-specific response to postprandial lipemia, unsupported by data obtained in the cohorts analyzed for heritability.

---

## Round 0.2 · accepted · Accept

Thanks for addressing these points. I am delighted to accept this version and look forward to seeing it in print.